# Model-based test case prioritization using selective and even-spread count-based methods with scrutinized ordering criterion

**Muhammad Luqman Mohd-Shafie**, **Wan Mohd Nasir Wan-Kadir***,
**Muhammad Khatibsyarbini, Mohd Adham Isa**

Department of Software Engineering, School of Computing, Faculty of Engineering, Universiti Teknologi Malaysia, Johor Bahru, Johor, Malaysia

* wnasir@utm.my

**Data Availability Statement:** All data files for replication can be downloaded from: https://doi.org/10.7910/DVN/5LZ51B.

## Abstract

Regression testing is crucial in ensuring that modifications made did not introduce any adverse effect on the software being modified. However, regression testing suffers from execution cost and time consumption problems. Test case prioritization (TCP) is one of the techniques used to overcome these issues by re-ordering test cases based on their priorities. Model-based TCP (MB-TCP) is an approach in TCP where the software models are manipulated to perform prioritization. The issue with MB-TCP is that most of the existing approaches do not provide satisfactory faults detection capability. Besides, their granularity of test selection criteria is not very good and this can affect prioritization effectiveness. This study proposes an MB-TCP approach that can improve the faults detection performance of regression testing. It combines the implementation of two existing approaches from the literature while incorporating an additional ordering criterion to boost prioritization efficacy. A detailed empirical study is conducted with the aim to evaluate and compare the performance of the proposed approach with the selected existing approaches from the literature using the average of the percentage of faults detected (APFD) metric. Three web applications were used as the objects of study to obtain the required test suites that contained the tests to be prioritized. From the result obtained, the proposed approach yields the highest APFD values over other existing approaches which are 91%, 86% and 91% respectively for the three web applications. These higher APFD values signify that the proposed approach is very effective in revealing faults early during testing. They also show that the proposed approach can improve the faults detection performance of regression testing.

## 1.0 Introduction

Regression testing assures that the changes made to a particular software system did not produce any adverse impacts on the software [1]. Unfortunately, regression testing suffers from several notable issues. One of them is the execution cost. Regression testing is among the costliest phases in the software development life cycle [2]. About 80 percent of the testing budget is

**Funding:** The authors would like to express their deepest gratitude to Research Management Center (RMC), Universiti Teknologi Malaysia (UTM) and Ministry of Higher Education Malaysia (MOHE) for their financial support under the Research University Grant Scheme (Vot number Q. J130000.2516.19H64).

**Competing interests:** The authors have declared that no competing interests exist.

spent on it [3]. According to a statistic by Memon, Gao [4], every day at Google, 150 million tests are run on more than 13 thousand projects that require 800 thousand builds. In this circumstance, even with modern test frameworks that re-run predefined tests, the execution cost will still be unbearable if the entire test suite needs to be executed. Another problem in regression testing is regarding time consumption. According to Elbaum, Kallakuri [5], a report of an industrial collaborator stated that one of its products containing 20,000 lines of code requires seven weeks for the entire test suite to be carried out. These issues will surely prevent regression testing from running effectively. One of the side effects is in terms of performance of faults detected. Consider a situation where regression testing needs to be halted abruptly because of cost or deadline issue; significant faults located in the neglected test cases will be left undetected.

Various solutions have been proposed by researchers to overcome the issues mentioned earlier. Yoo and Harman [1] classified the approaches that can increase the effectiveness and efficiency of regression testing into three main categories. They are test suite minimization (TSM), test case selection (TCS) and test case prioritization (TCP). Approaches in TSM remove any obsolete or unessential test cases permanently from the test suite [6]. TCS approaches pick relevant test cases from the test suite according to certain criteria [7]. Last but not least, TCP approaches reorganize test cases from the original test suite into a prioritized test suite. The prioritization is done according to a specified purpose given that the test cases that contribute the most to the purpose are given the highest priorities [8].

There are several reasons why an approach based on TCP is proposed in this study over other categories mentioned earlier. TSM, although saves a lot of cost by reducing test suite size, possesses a risk where significant test cases that reveal faults might be permanently eliminated [1, 2]. This issue can surely compromise the fault detection capability of a test suite. TCS on the other hand only selects necessary test cases without removing the needless ones. However, TCS also suffers the same issue as TCM where prominent test cases that reveal faults might be omitted [2]. Using TCP, not a single test case is neglected and test cases that have the possibilities of addressing the specified objective are prioritized first. This technique is advantageous because if testing needs to be stopped, at least the most important test cases have been executed.

One way of categorizing TCP approaches is to divide them into code-based and model-based approaches [9]. The vast majority of TCP approaches are code-based. Although very popular, code-based TCP exhibits several disadvantages that will be discussed in Section 2. MB-TCP was first proposed by Korel, Tahat [10]. In their study, the system models are utilized to prioritize test cases rather than the source code. One of the reasons why an MB-TCP approach is proposed in this study is because of cheaper execution cost over the code-based approach [10].

The aim of this study is to propose an MB-TCP approach that can improve the effectiveness of regression testing in terms of performance of fault detection. This approach is called Enhanced Model-based Prioritization using **S**elective and **E**ven-spread Count-based Methods Combination with **S**crutinized **O**rdering **C**riterion (SESOC). SESOC improves the existing approaches in terms of fault detection performance by combining several of those approaches while adding a new ordering criterion to increase the granularity of test selection criteria. The finite state machine (FSM) is utilized as the system model to be manipulated for prioritization in this study. The prioritization effectiveness of SESOC is evaluated by benchmarking it with the performance of the existing MB-TCP approaches in the literature that utilized FSM as the model. For this purpose, the average of the percentage of faults detected (APFD) metric is used. This study is an extension of the previous work done by Shafie and Kadir [11]. In this present study, the proposed approach is refined by providing a more comprehensive

elaboration. More objects of study are also included so that better insights can be obtained about the effectiveness of the proposed approach.

In summary, there are three contributions to this study. The first one is an enhancement of existing MB-TCP approaches using FSM. The existing MB-TCP approaches using FSM, selective test prioritization (STP) proposed by Korel, Tahat [10] and the even-spread count-based test prioritization proposed by Tahat, Korel [12], are enhanced by combining their implementations while also adding a new criterion to increase the granularity of selection criteria hence boosting the performance of prioritization. Secondly, a new prioritization criterion is introduced called the degree of code changes. Using this criterion, the priority of each test case will be more detailed, thus, enabling the significance of each test to be further analyzed. The third contribution is a detailed experiment that is conducted to evaluate and compare the performance of the proposed approach with selected existing approaches in the literature.

The remainder of this paper is organized as follows. In Section 2, a comprehensive elaboration of each related domain is shown. Then, Section 3 shows the related works in MB-TCP. Next, a detailed elaboration regarding SESOC is presented in Section 4. After that, the details of the conducted experiment are shown in Section 5. Lastly, the conclusion and future recommendations are discussed in Section 6.

## 2.0 Background

In this section, further clarifications regarding MBT, FSM, and MB-TCP are presented.

### 2.1 Model-based testing

The aim of testing in the context of software engineering is to show whether the behaviours of an actual software system is the same as the expected behaviours or vice versa [13]. Generally speaking, fault detection is the main goal of testing which is done by searching dissimilarities between the actual and the planned behaviours of the system under test (SUT), as indicated by its requirements. Shafique and Labiche [14] stated that software testing by spotting its executions on valued inputs is probably the most commonly used verification technique in the evaluation of an SUT. MBT is a branch of software testing under black-box testing that relies on the behaviour models that visualize the expected behaviours of a SUT. In other words, the test oracle problem is addressed in MBT by constructing the test oracle using the behaviour models [15]. MBT extends testing automation from test design to test execution by making use of automatic test generation and execution from the model [16]. MBT can save testing cost because SUT behaviour models are utilized for automatic test case generation, unlike conventional testing where each test case must be coded by the test engineer [17]. In addition, these generated test cases can be executed automatically using a test automation tool to alleviate the human oracle cost problem. Most approaches in MB-TCP generate test cases using the same procedure as MBT. Therefore, it is crucial to comprehend the MBT process.

The process of MBT explained here is referred from Utting, Pretschner [13]. Firstly, test models are built from the specification documents or informal requirements of the SUT by the test engineer. How they are created depends on the type of SUT. For this particular study, web application (web app) is the SUT and the way it is modelled using FSM is explained in Section 5.1. It is important to note that the test models must be simpler (more abstract) compared to the SUT which mean they are easier to inspect, change and retain. Otherwise, the effort to validate the models would be equivalent to validating the SUT itself.

FSM is commonly used to model system behaviours in MBT and is utilized in this research. Javed, Minhas [18] conducted a systematic literature review (SLR) study of MBT for web app and they concluded that majority of the identified MBT approaches are based on FSM and

several other behavioural models like activity diagram, state diagram and extended FSM (EFSM). Furthermore, Sabbaghi and Keyvanpour [19] conducted a study to review the most used state-based models in MBT. They also stated that FSM is widely used in MBT alongside EFSM, state diagram, timed automata and Markov usage model. These review studies proved that the FSM model is frequently used in performing MBT. FSM is a good choice for performing testing and prioritization because it can be exploited to generate abstract test cases which later can be run on the actual system during testing. Although this model is an abstraction of the system itself, crucial details are not abstracted out which makes it executable on its own. Therefore, this model is sufficiently precise to be used as a foundation for generating good abstract test cases. The next section presents a detailed explanation of this model.

The second step in MBT is to decide on the test selection criteria. This step is done to drive the automatic test case generation so a good quality test suite can be generated. After deciding the criteria of test selection, they are transformed into test case specifications in the third step. Test case specifications formalize the notion of test selection criteria and make them operational. The fourth step in MBT is where a set of abstract test cases is generated, which aims to satisfy all of the test case specifications. An automatic test case generator is utilized in this step to develop a test suite given the models and the test case specifications.

Finally, the test suite is ready to be run in the fifth step. This step is done manually by a person or by a test execution environment that supports the ability to execute the tests and record their verdicts automatically. During the execution process, test inputs are first concretized and then sent to the SUT. Next, the resulting concrete outputs from the SUT are abstracted to obtain the high-level actual results. These actual results will be compared with the expected results or in this case, the abstract test cases, to determine their verdicts. These concretization and abstraction processes are handled by a component called an adapter. Fig 1 illustrates the overall process of MBT with the corresponding steps labelled.

Nevertheless, there are some limitations in using MBT approach for testing. One of them is that the success of testing depends on the quality of the artefacts used for creating the test model [20]. This means that if incomplete or incorrect software artefacts are used, their defects will propagate to the constructed test model and affect the generated test cases. However, this issue could be countered because the test model illustrates the behaviours of the software system better compared to the artefacts like specification documents or informal requirements. Therefore, if the test model is validated first, the defects introduced in the software artefacts could be identified and fixed before they spread to the testing phase. Another issue that needs to be considered when using MBT is about the cost. This cost includes the knowledge and expertise needed to create and maintain the test model and also to implement the automation of test case generation and execution. However, these expenses are mostly needed during the early phase of testing. When the recurring maintenance phase begins, the full benefits of MBT will be utilized.

## 2.2 Finite state machine

The finite state machine (FSM) is a model of computation. It can change from one state to another given that there are some external inputs which it can respond to. This model is usually applied to replicate sequential logic in computer programs. It is utilized in many modern machines that carry out fixed order of actions depending on the sequence of responses provided. Some examples of these machines are vending machine, combination lock and elevator.

The formal definition of FSM is a 5-tuple $S = (S,I,O,h_s,s_0)$, where $S$ denotes a nonempty finite set of states with the designated initial state $s_0$, $I$ and $O$ denote the nonempty finite *input* and *output* alphabets respectively, and $h_s \subseteq I \times S \times S \times O$ is a *transition* relation [21]. An FSM can

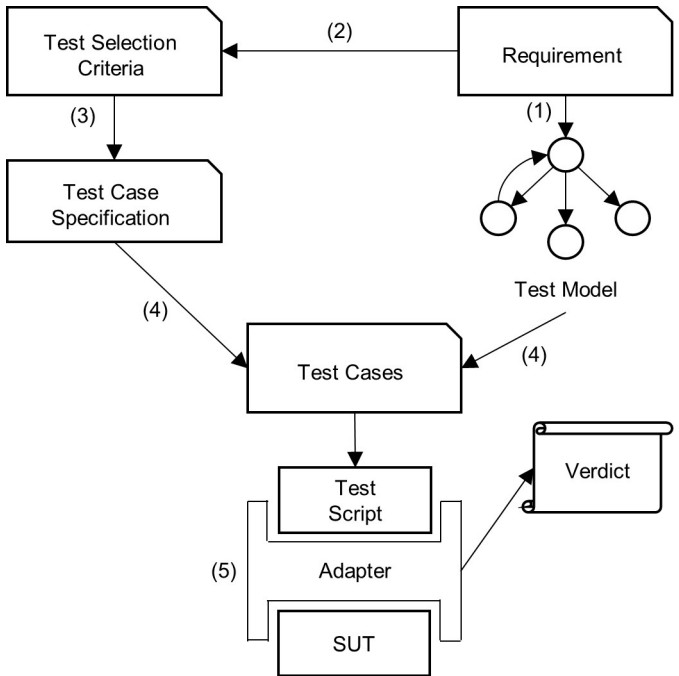

**Fig 1. MBT overall process (adapted from Utting, Pretschner [13]).**

be illustrated using a directed graph called a state diagram. States are depicted by vertexes (node) with the edges (arrows) as the transitions that connect between two vertexes. Each edge is associated with a specific input which when triggered, changes the machine from its current state to the next one set by that edge. Fig 2 shows the FSM model for an Online Jewellery Store which is later used in the experimentation.

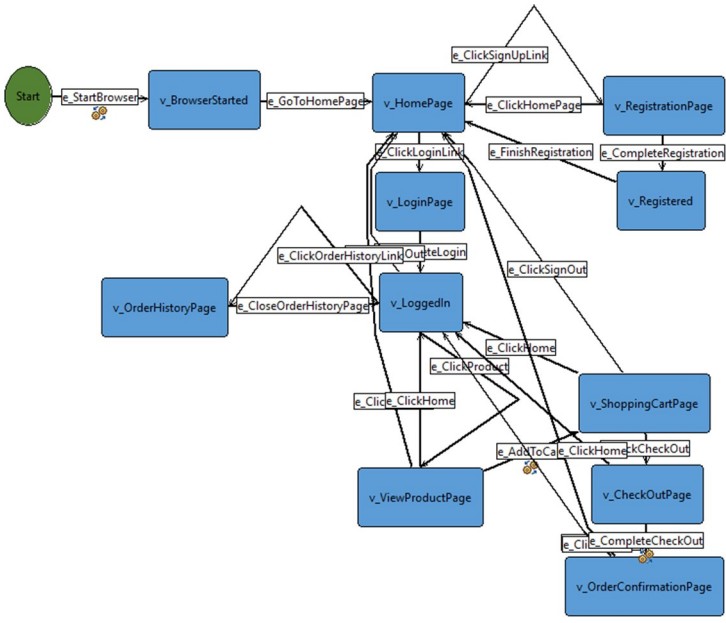

**Fig 2. FSM model of Online Jewellery Store.**

## 2.3 Model-based test case prioritization

Test case prioritization (TCP) is a regression testing technique that re-orders test cases from the original test suite according to a specified objective where test cases that most serve the objective are given the highest priorities [22]. The interpretation of the TCP problem suggested by Elbaum, Malishevsky [23] is considered in this study:

Given: $T$, a test suite; $PT$, the set of permutations of $T$; $f$, a function from $PT$ to the real number. Problem: Find $T' \in PT$ such that

$$(\forall T'')(T'' \in PT)(T'' \neq T')[f(T') \geq f(T'')] \tag{1}$$

In this interpretation, $PT$ serves as the set of all anticipated orderings of $T$, while $f$ is the function when implemented to any of the ordering, creates an *award value* for it. In short, the interpretation expects that higher award values are preferable than the lower ones. The function $f$ is the most crucial part that represents the approach used to prioritize test cases. There are a number of possible goals when referring to prioritization in this context. Elbaum, Malishevsky [23] also stated some of the goals in their study which are:

- To increase the rate of early fault detection when executing the test suite.

- To escalate the code coverage under test at a faster pace when executing the test suite.

- To boost the credence in the system's reliability at a faster rate.

- To increase the possibility of identifying faults associated with a particular code modification quicker in the testing process.

In this study, only the "rate of early fault detection when executing the test suite" objective is focused.

Over time, numerous approaches for TCP have been proposed. One way of categorizing the existing TCP approaches is to divide them into code-based and model-based types [9]. In code-based TCP, prioritization is done by utilizing the source code information. Most of the TCP approaches proposed in the literature are code-based. From an SLR conducted by Khatibsyarbini, Isa [24], only five percent out of 80 studies collected regarding TCP are model-based approaches. Furthermore, Catal and Mishra [2] found out that the most investigated prioritization method was coverage-based which is 40 percent of all the various approaches they had gathered. Coverage-based is another term for code-based TCP where the more code coverage of the software system is achieved by a test suite, the more chances faults can be revealed earlier during testing. The downside of code-based prioritization is the requirement of code knowledge to prioritize test cases which means prioritization cannot begin until the source code is available. Another drawback is that most of them are language dependent so the testing process will become troublesome in cases where the program is written in diverse programming languages [25].

On the other hand, MB-TCP exploits the models of the software system to carry out prioritization. Generally, any kind of TCP approach that manipulates the system models in its implementation can be categorized as MB-TCP. Some examples of system model are use case diagram, sequence diagram, state machine diagram and activity diagram. The key advantage of MB-TCP is that execution of the system models is quicker compared to the execution of the system code itself during testing [10]. Because of the high abstraction level of system models, capturing the system's behaviours and structures are less complicated compared to using the source code [12]. Therefore, MB-TCP is considerably inexpensive, resource-wise and time-wise compared to code-based TCP [10]. Despite that, MB-TCP also has its own weaknesses. One of the major flaws is its dependence on the correctness and completeness of the system

models [26]. More limitations are shown when the related approaches in MB-TCP are discussed in Section 3.

**2.3.1 Model-based test case prioritization using finite state machine.** MB-TCP approaches using FSM model in this study including the proposed approach are for a class of modifications where only the source code of the SUT is altered and not the models. For such class of modifications, there will be no distinction between two versions of the model because they are not changed. This circumstance usually happens because many changes required in a software system are caused by insignificant bugs or technical glitches. These changes do not require modifications in the model because the model only focuses on the behaviours and structures of a software system while abstracting out the underlying processes and details related in the source code [12]. Modifications in the data structure and enhancing the efficiency of the coding are some examples that do not require modifying the models.

When these modifications are done to the coding, the developers identify the transitions in the model that are affected by them. These transitions are called modified transitions. It is fairly straightforward to recognize modified transitions that are associated with source code modifications. This is because model transitions are usually translated into functions in the source code [27]. Therefore, if any modification is made in a function, all transitions that link to this function will be labelled as modified transitions. The identification of these modified transitions in the model is very important because this information will be utilized during the prioritization process.

## 3.0 Related works

In this section, several existing MB-TCP approaches in the literature with their identified limitations are discussed. Al-Herz and Ahmed [28] proposed an approach named Degree Measure Method (DMM). This approach utilizes the Object Relation Diagram (ORD) model which represents the design structure of a web app. DMM ranks components according to their fan-in degree, then prioritizes test cases based on the rank of components. In this context, fan in degree means the number of components that lead to the component in consideration. The logic behind this approach is that most of the other components will fail to get the required services if this high fan-in degree component malfunctions [28]. The weakness of this approach is the assignment of priority when two components have the same fan-in degree. The possibility of situations where two components might get similar priorities is high because the granularity of test selection criteria is quite low in this approach. Their proposition to address this drawback is by assigning additional criteria such as component type and fan-in edge type.

Another approach in MB-TCP is proposed by Sapna and Mohanty [26] that uses the structural aspects of the use case diagram and activity diagram. In their approach, both diagrams are used as the input for prioritization. The process starts with capturing data from all use case diagrams to calculate use case priority. Next, scenarios are extracted from activity diagrams and weights are assigned to their nodes and edges. The weight of path (scenario) is calculated and finally prioritized by totalling the sum of the priorities starting at level 1 of the schema and moving down by adding the weights of all the nodes up to the scenario weight. The downside in this approach is its dependence on the correctness and completeness of the use case diagram and activity diagram. For example, if the activity diagram is not complete, there will be possibilities where some requirements are not captured. As a result, the scenarios will not be generated and this will affect the overall prioritization.

Furthermore, an approach called model dependence-based test prioritization was proposed by Korel, Tahat [10] that utilizes FSM to perform prioritization. This approach was elaborated

by them in further details in their extended version of studies for modification made both in the software system and models and for modification for which models are not modified (only source code is modified) [12, 29]. Concisely, this approach does model dependence analysis to determine the patterns of how added and deleted transitions connect with the modified model and lastly utilizes this information to prioritize test cases. Despite that, this approach increases execution time because it gathers extra information and needs more analysis compared to other approaches proposed by them. Furthermore, the whole model execution trace must be stored to compute the interaction patterns, thus raising resources usage.

Another example of MB-TCP approach that uses FSM model is STP [10]. In this approach, high priorities are given randomly to tests that execute modified transitions in the model. The limitation in this approach is that only prioritizing test cases based on their number of modified transitions randomly is insufficient and does not have a significant impact on improving fault detection. Another example of an FSM based approach is basic frequency-based prioritization (BFP). In this approach, the frequency of modified transitions traversed by a test case is considered. Tests that traverse greater frequency of modified transitions will be assigned higher priorities compared to tests that traverse lower frequency of modified transitions. The drawback of this approach is that modified transition frequencies of test cases are not a good type of information to be used for prioritization.

In addition to that, there is also an FSM based approach called even-spread count-based test prioritization [12]. In this approach, all modified transitions are given the same chances to be covered during testing. This approach can provide a good prioritization result, however, it can still be improved so that better prioritization result can be generated. Many MB-TCP approaches using FSM also did not have a good granularity for selecting test cases to be prioritized. This low level of granularity will prevent these approaches from getting good prioritization results. These limitations in MB-TCP approaches that utilize FSM show that refinement can be added to improve the effectiveness of early fault detection. These limitations also motivate the current study to propose an approach that can improve the prioritization effectiveness of the existing MB-TCP approaches using FSM.

## 4.0 Proposed approach

In this section, the proposed MB-TCP approach which is called SESOC is explained in detail. Firstly, the overview of SESOC is presented. Then, the implementation process is explained. Lastly, an example implementation of SESOC in prioritizing test cases is demonstrated.

### 4.1 Overview

The proposed approach is a combination of the STP approach and even-spread count-based test prioritization approach proposed by Korel, Tahat [10] and Tahat, Korel [12]. In addition, a newly introduced criterion is applied to further scrutinize the prioritization of test cases. This approach also tries to overcome the downsides mentioned in the related MB-TCP approaches in the earlier section. Firstly, to provide a model that can support correct and complete traceability to the system, the FSM is chosen. This model is advantageous because it gives an understandable visual representation of various types of behaviour associated with transitions. It is also utilized numerously to illustrate systems at a higher level of abstraction for better comprehension and traceability. Secondly, an additional criterion is introduced which is called the degree of code changes to increase the granularity of test selection criteria and prevent the confrontation of situations where two nodes having the same degree of importance. Lastly, this proposed approach will be both simple and comprehensible yet providing a solid and reliable prioritization result.

The idea of SESOC is that higher priorities are assigned to tests that execute more modified transitions in the model while balancing the number of executions of modified transitions during testing. A modified transition will also be assigned a degree of code changes and the higher the degree of code changes of a transition, the higher its priority will be. Ultimately, tests with modified transitions that have high degree of code changes executed the least number of times will be given higher priorities. Based on the explanation above, the approach is named Enhanced Model-based Prioritization using Selective and Even-spread Count-based Methods Combination with Scrutinized Ordering Criterion or SESOC for short. As stated earlier in Section 2.3.1, SESOC is utilized for a class of modifications where only the source code is changed and the models stay unchanged. Therefore, this approach is appropriate to be used for finding bugs that might be introduced during the maintenance phase of a software development life cycle.

## 4.2 Process

Fig 3 depicts the framework of the proposed SESOC approach. At first, two inputs are required. They are the original test suite generated from the MBT process and the modified transition information. Using these two inputs and the introduced degree of code changes, the transition scores for all transitions are calculated. This process is the first process depicted as P1 in Fig 3. Then, the second process, represented as P2 in Fig 3, is done. In this particular process, test case score for each test case is calculated. Then, the test cases are ranked and divided

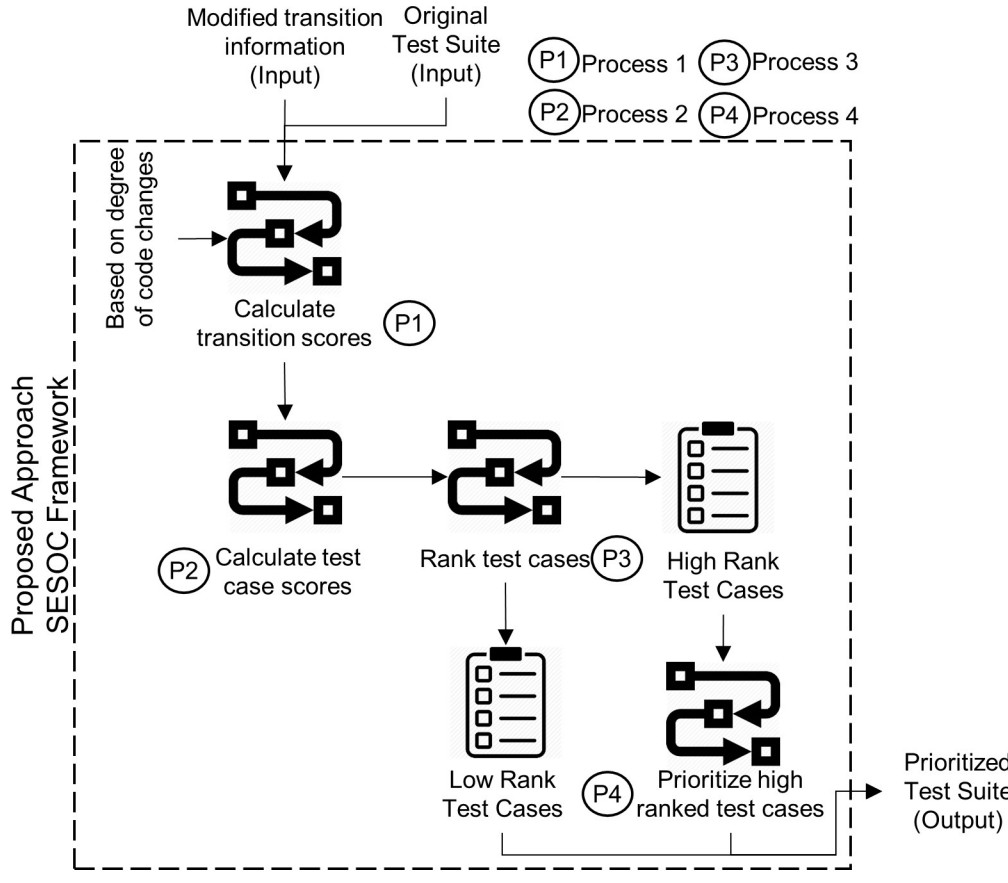

**Fig 3. SESOC framework.**

into high-rank and low-rank test cases in the third process represented as P3 in Fig 3. Next, the high-rank test cases are further scrutinized for high priority test cases to increase the granularity of test selection criteria, represented as P4 in Fig 3. After that, the prioritized high-rank test cases are combined back with the low-rank test cases to obtain the prioritized test suite. This prioritized test suite is the output of SESOC implementation. The next paragraph discusses the implementation of SESOC in further detail.

In the first process, represented by P1 in Fig 3, each modified transition $T_j \in MT$ will be given a transition score, $ScT(T_j)$ based on its degree of code changes, where $T$ is a transition, $j$ is the modified transition number and $MT$ is a set of all modified transitions. This is the first process in the framework. Degree of code changes is calculated by identifying how many lines of source code associated with a transition that are modified and assigning the number of modified lines as the score for that transition. In this context, a modification includes addition, deletion and change. Recall that in the earlier explanation about MB-TCP using FSM in Section 2.3.1, if any modification is made in a function of coding, all transitions that refer to this function will be labelled as modified transitions. For example, suppose in a function there are two lines of code where modifications are made, and the transition associated with that function is $T_1$, then $ScT(T_1) = 2$. For transitions that are not modified, their scores will simply be zero.

The second step, depicted by P2 in Fig 3, is to calculate the test case score, $Sct(t_i)$ by summing all the scores of modified transitions traversed by that test case. This is the second process in the framework. Eq 2 shows the calculation to determine a test case score.

$$Sct(t_i) = \sum_{j=1}^{J} ScT(T_j), T_j \in A(t_i) \tag{2}$$

where $A(t_i)$ is a set of modified transitions executed by a test $t_i$, $i$ is the test case number, $j$ is the modified transition number and $J$ is the total number of modified transitions traversed by test case $t_i$. After the scores for all test cases have been calculated, the test cases with zero $Sct(t_i)$ are isolated and treated as low-rank test cases because they do not traverse any modified transition. This is because faults are unlikely to be located at the functions that are not modified. The other test cases with non-zero $Sct(t_i)$ are treated as high-rank test cases. This isolation of high and low-rank test cases is the third process in the framework shown by P3 in Fig 3.

Then, the fourth process, represented by P1 in Fig 3, is done. The test case in high rank with the highest $Sct(t_i)$ is appended into the last position of the prioritized test suite, $TS_P$. In an unlikely event of more than one test case having the highest score, one is randomly chosen between them. Then, a set $E$ that contains the modified transitions that have been appended into $TS_P$ is determined. After that, $Sct(t_i)$ for each test case is updated. For updating the test case score, if a test case in high rank contains the modified transitions in set $E$, then the transition scores of those modified transitions in the test case will be eliminated. For example, initially $A(t_6) = \{T_1, T_2\}, ScT(T_1) = 2$ and $ScT(T_2) = 1$, therefore $Sct(t_6) = 3$. Assuming that another test case is appended to $TS_P$ and set $E$ is updated where $E = \{T_1\}$, the updated score will be $ScT(t_6) = 1$.

Subsequently, the remaining test cases in high rank are checked if all of their $Sct(t_i) = 0$. If yes, all of them are appended randomly into the last position of prioritized $TS_P$. If no, then the next test case in high rank with the highest $Sct(t_i)$ is put into $TS_P$ same as the first step in the fourth process until the updating and checking steps. The process continues looping until all test cases in high rank are ordered in $TS_P$. Lastly, test cases in low rank that traverse no modified transition are ordered randomly at the end of $TS_P$. Fig 4, quoted and altered from the previous work of Shafie and Kadir [11], shows the flowchart of the SESOC algorithm. P1, P2, P3

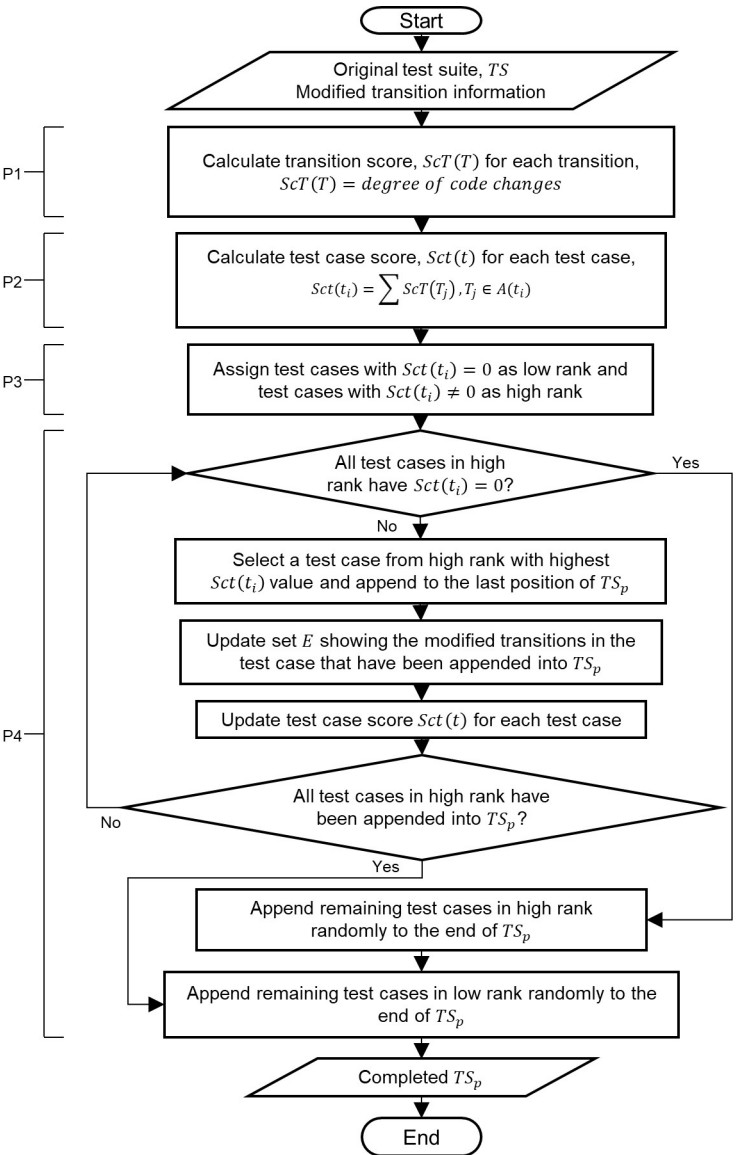

**Fig 4. Flowchart of SESOC (quoted and altered from Shafie and Kadir [11]).**

and P4 at the left-hand side reflect the corresponding processes that are in parallel with the processes in the framework shown in Fig 3.

## 4.3 Example

Suppose an example where $MT = \{T_1, T_2, T_3, T_4, T_5\}$, $TS = \langle t_1, t_2, t_3, t_4, t_5, t_6, t_7, t_8, t_9, t_{10} \rangle$ where $TS$ is a test suite containing 10 test cases and $A(t_1) = \{T_1, T_2, T_3\}$; $A(t_2) = \{T_3, T_4, T_5\}$; $A(t_3) = \{T_3, T_4\}$; $A(t_4) = \{T_5\}$; $A(t_5) = \{T_1\}$; $A(t_6) = \{T_1, T_2\}$; $A(t_7) = \{T_2, T_4\}$; $A(t_8) = \{T_2, T_3, T_4\}$; $A(t_9) = \emptyset$; $A(t_{10}) = \emptyset$. Assuming that the transition score $ScT(T_j)$ for each modified transition is calculated where $ScT(T_1) = 2$; $ScT(T_2) = 1$; $ScT(T_3) = 1$; $ScT(T_4) = 1$; $ScT(T_5) = 3$, then the test case score $Sct(t_i)$ for each test case is calculated where $Sct(t_1) = 4$; $Sct(t_2) = 5$; $Sct(t_3) = 2$; $Sct(t_4) = 3$; $Sct(t_5) = 2$; $Sct(t_6) = 3$; $Sct(t_7) = 2$; $Sct(t_8) = 3$; $Sct(t_9) = 0$; $Sct(t_{10}) = 0$.

Based on the $Sct(t)$ values, it can be observed that $Sct(t_2)$ has the highest value, therefore, it will be appended first, $TS_P = \langle t_2 \rangle$. Next, the set $E$ where modified transitions that have been appended into $TS_P$ is determined where $E = \{T_3, T_4, T_5\}$. Then, the test case score $Sct(t_i)$ for each test case is updated where $Sct(t_1) = 3$; $Sct(t_2) = 0$; $Sct(t_3) = 0$; $Sct(t_4) = 0$; $Sct(t_5) = 2$; $Sct(t_6) = 3$; $Sct(t_7) = 1$; $Sct(t_8) = 1$; $Sct(t_9) = 0$; $Sct(t_{10}) = 0$. Based on the updated $Sct(t_i)$ values, it can be observed that $Sct(t_1)$ and $Sct(t_6)$ have the highest value. Therefore, one random test case between these two is appended into $TS_P$ and assuming that $t_1$ is chosen, then $TS_P = \langle t_2, t_1 \rangle$. This event is very unlikely to happen in a real-world situation considering the complexity and size of the test suite. Then, set $E$ will be updated where $E = \{T_3, T_4, T_5, T_1, T_2\}$. The updated test case score will be $Sct(t_1) = 0$; $Sct(t_2) = 0$; $Sct(t_3) = 0$; $Sct(t_4) = 0$; $Sct(t_5) = 0$; $Sct(t_6) = 0$; $Sct(t_7) = 0$; $Sct(t_8) = 0$; $Sct(t_9) = 0$; $Sct(t_{10}) = 0$. Considering that all test cases scores are 0, they are appended randomly in $TS_P$. The remaining test cases which traverse no modified transition in $TS$ will be selected randomly to be appended in $TS_P$. Therefore, a possible prioritized test suite will be $TS_P = \langle t_2, t_1, t_3, t_4, t_5, t_6, t_7, t_8, t_9, t_{10} \rangle$.

## 5.0 Experiment framework

In this section, an experiment is conducted with the aims to evaluate and compare the effectiveness of early fault detection of SESOC with the selected existing approaches in the literature. The scope of this experiment is determined by describing its objective. The template for goal definition is followed by the one that was originally presented by Basili and Rombach [30]. A more detailed description of this template can be found in Wohlin, Runeson [31]. For this experiment, the goal summary is shown below:

Analyze *the MB-TCP approaches using FSM*

for the purpose of *evaluation*

with respect to *effectiveness in prioritizing fault detecting tests*

from the point of view of *the software tester*

in the context of *web app testing.*

### 5.1 Objects of study

Web apps are utilized as objects of study in this experiment. Therefore, the justifications of their usage are presented in this section. Nowadays, modern web apps are intricate and highly interactive. They have complicated interfaces and various back-end software elements that are integrated in many ways [32]. Web-based systems also tend to scale rapidly and go through frequent alterations because of new technological opportunities and users feedbacks [33]. Because of these circumstances, the iterative development process based on continuous changes and rapid prototyping is a very good choice for web app development [34]. Nevertheless, cost issue could occur during the regression testing because of the continuous testing in this development model. Therefore, TCP is a suitable technique to be applied here because it prioritizes faults revealing test cases thus increasing efficiency in this rapid development environment.

Secondly, web apps are utilized because they can be modelled using FSM by representing them in the form of states and transitions between states [32]. States can be associated with page validations where the user is currently browsing. On the other hand, a transition can be associated with clicking buttons, entering texts or whatsoever actions, that when triggered by

the user, changes the state of the web app. Therefore, MB-TCP using FSM is suitable for web app testing and deduces the reasons why web apps are chosen in this study.

In MB-TCP, the FSM model of a software system is required to generate the test cases for testing purpose. The software system itself that contains faults is also required to measure the approaches' prioritization effectiveness. Unfortunately, system models for real-world commercial software with their respective software systems are not available freely [12, 29], and not to mention that this study requires an FSM model specifically. In addition, most of the MB-TCP studies in the literature created their own system models for testing and did not make them available to the public domain for other researchers to utilize as datasets.

Consequently, this study made use of web apps that are available in the public domain as the objects of study. The FSM models for these web apps are created during the experiment. Three web apps are obtained from Sanjeev [35]. They are Online Jewellery Shopping, Car Rental System and Blood Bank Management System. These three are selected because they represent web apps of different sizes. From this difference, the effect on the prioritization result can be analyzed. All of them were included with their essential files and databases so that they can be executed in localhost for testing purpose. Based on their interfaces and functionalities, these open source web apps are similar and reflect those web apps from actual industrial use.

Table 1 shows the characteristics of the selected web apps that are of interest in this study. The number of states and number of unique transitions refer to those in the FCM models of these three web apps. From these numbers, it can be observed that the three models have different sizes. Only unique transitions are counted in because there are many similar transitions that are used by different states. These similar transitions will execute the same functions in the implementation so the effect of modifications in a function will affect the similar transitions equivalently. Therefore, these similar transitions are treated as only one transition. The last column refers to the lines of code (LOC) in all project files for each web app excluding blank and comment lines, calculated using the *cloc* tool [36]. More information regarding the models and the source code are available in the repository [37].

## 5.2 Independent variable

The independent variable is the MB-TCP approach using FSM with six treatments consisting of five distinct approaches from Korel, Koutsogiannakis [27], Tahat, Korel [12] and SESOC itself. The five approaches are STP, basic count-based prioritization (BCP), round-robin count-based prioritization (RCP), basic frequency-based prioritization (BFP) and round-robin frequency-based prioritization (RFP). A brief explanation of these five approaches is provided. They were selected based on our latest SLR work that reviewed existing MB-TCP approaches [8]. To ensure a fair comparison, only approaches that utilized theh FSM model were chosen. Approaches that utilized other types of model implemented different prioritization criteria depending on the information provided from the model. For that reason, comparison with them was not made because it could affect the internal validity of the experiment.

In STP, high priorities are given to tests that executed modified transitions in the model, meaning that if a test traverses at least one modified transition in its execution, that test will be

**Table 1. Characteristics of the selected web apps.**

| Web App | Number of Web Pages | Number of States (FCM model) | Number of Unique Transitions (FCM model) | LOC |
|---|---|---|---|---|
| Online Jewellery Shop | 14 | 11 | 16 | 22684 |
| Car Rental System | 13 | 16 | 28 | 24668 |
| Blood Bank Management System | 16 | 19 | 32 | 28663 |

given high priority. Otherwise, low priorities are given to tests that traverse no modified transition in their execution. The order of tests for both high priority and low priority groups in $TS_P$ is done randomly.

For BCP, the main information used is the number of unique modified transitions traversed by a test. The idea is that tests that traverse a greater amount of unique modified transitions will be assigned higher priorities compared to tests that traverse lower amount of unique modified transitions. RCP is quite similar to BCP but with a slight distinction. The fundamental of this approach is that using the BCP approach, one test case is picked from the first priority group with the most unique modified transitions, then another one is picked from the second priority group with the second most unique modified transitions and so on. When the last priority group with only one unique modified transition has been reached, the selection goes back to the first priority group and continues again. This order of selection repeats until all tests that traversed at least one modified transition are picked. Only then the tests with no modified transition traversed are selected randomly into the end of $TS_P$.

For BFP, the main information used is the frequency of modified transitions traversed by a test. The difference between count-based and frequency-based is that count-based calculates the unique number of modified transitions in a test while frequency-based calculates the frequency of modified transitions in a test regardless of whether they are unique or similar modified transitions. The idea of this approach is similar to BCP. While BCP prioritizes test cases with the most unique number of modified transitions, BFP prioritizes test cases with the highest frequency of modified transitions. In RFP, the criteria for test cases selection will be based on RCP but the information used will be the frequency of modified transitions traversed by a test.

## 5.3 Dependent variable

One of the metrics that is popularly used to evaluate a TCP approach's effectiveness in early fault detection is the APFD. Rothermel, Untch [38] and Elbaum, Malishevsky [23] are some of the earliest studies that mentioned this evaluation metric. They defined APFD as a metric used to quantify how rapid a prioritized test suite locates faults. The value of APFD result can be from 0 to 1 where a bigger value shows greater fault detection rate. The equation for calculating the APFD value acquired from Elbaum, Malishevsky [39] is shown as follows:

$$APFD = 1 - \frac{TF_1 + TF_2 + \ldots + TF_m}{nm} + \frac{1}{2n} \tag{3}$$

Where $T$ represents a test suite consisting of $n$ test cases and $F$ is a group of $m$ faults discovered by $T$. $TF_i$ is the earliest test case in sequence $T'$ of $T$ which reveals fault $i$.

## 5.4 Experiment process

In this experiment, two tools were used to assist in running the experiment, namely GW4E and Selenium Webdriver. Both tools are based on the Java programming language. GW4E is the plugin of Graphwalker MBT tool in Eclipse Integrated Development Environment. It reads directed graphs and generates abstract test cases from them. Selenium Webdriver was utilized to automate web apps for testing purposes. It acted as an adapter that connected the abstract test cases generated from GW4E with the actual SUT. The function of the adapter has been explained in Section 2.1. Fig 5 illustrates the design of the experiment. The experiment process consists of model design, test selection criteria, test path generation, test execution and test prioritization. The explanation for each step is presented afterwards.

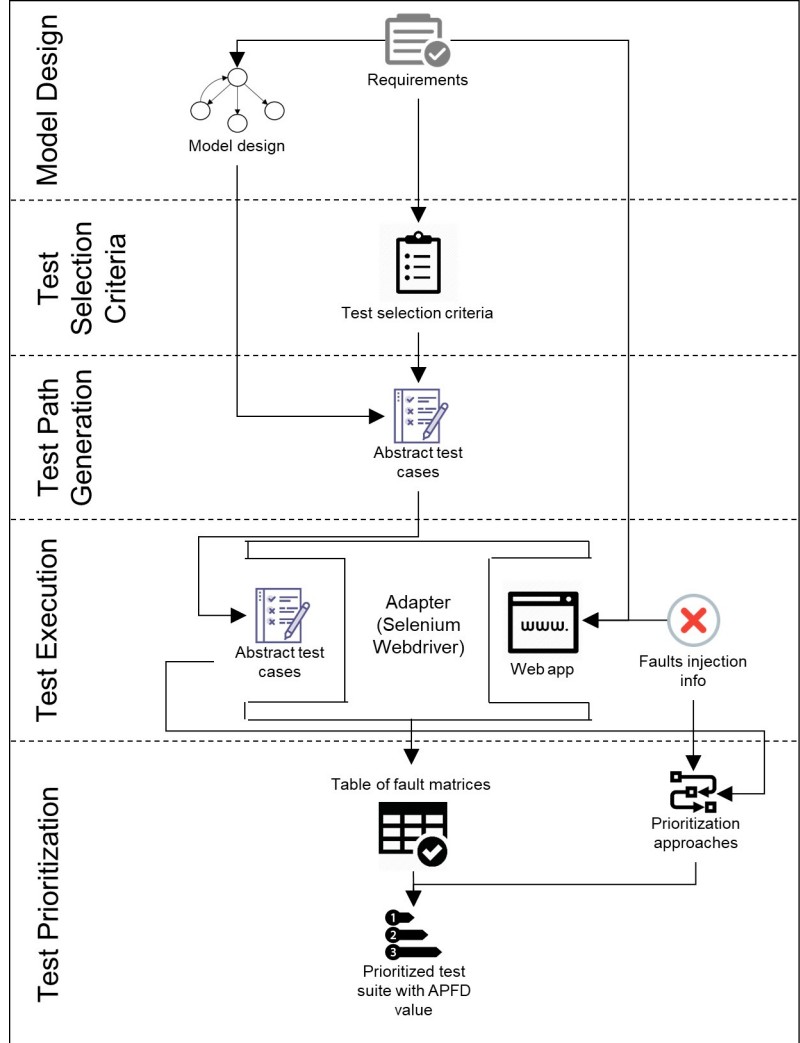

**Fig 5. Experiment design.**

Firstly, web apps were developed based on the requirements. In this experiment, web apps that were already developed with complete and working functionalities according to their requirements were acquired as stated earlier in Section 5.1. These web apps were modelled using FSM based on their requirements in GW4E. The FSM model for Online Jewellery Store is shown in Fig 2. The models for the remaining web apps are available in the repository [37].

Then, the criteria for test selection were decided to drive the automatic test generation. GW4E provided a special rule pattern for the test selection criteria consisting of a generator, a stop condition and a condition. For this experiment, it was required for the test suite to cover all the edges available in the model. Therefore, the pattern will be quick_random(edge_coverage(100)). quick_random means it will search the shortest paths through a model. edge_coverage means that the paths need to be covered are the edges. Lastly, 100 means that it needs to cover all 100 percent of the edges before stopping.

After model design and test selection criteria were done, the test paths generation can be executed. This experiment made use of offline path generation where the path generation was

```
@Override
public void v_HomePage() {
    waiter
        .until(ExpectedConditions
            .titleContains("BB Jewellery Online Store"));
    String message = waiter
                        .until(ExpectedConditions
                            .visibilityOfElementLocated(By
                                .xpath("//*[@id='welcome']")))
                        .getText();
    collector.checkThat(message, CoreMatchers.containsString("Welcome visitor"));
}

@Override
public void e_ClickSignUpLink() {
    waiter
        .until(ExpectedConditions
            .elementToBeClickable(By
                .xpath("/html/body/div[3]/header/div[1]/div[2]/a[2]")));
    waiter
        .until(ExpectedConditions
            .elementToBeClickable(By
                .xpath("/html/body/div[3]/header/div[1]/div[2]/a[2]")))
        .click();
}
```

**Fig 6. Implementation snippet of edges and vertexes using Selenium Webdriver for Online Jewellery Shopping.**

done once and stored in some intermediate format. Subsequently, GW4E utilized the created FSM model and the defined test selection criteria to generate the test path.

Next, Selenium Webdriver was used to connect the abstract test cases with the web app. Fig 6 shows an implementation snippet of edges and vertexes using Selenium Webdriver for Online Jewellery Shopping. In v_HomePage vertex, a validation was done to check whether the current page was really the homepage of Online Jewellery Shopping or vice versa. In e_ClickSignUpLink edge, the command instructed Selenium Webdriver to click the sign-up link. All the source code for the Selenium Webdriver implementation is available in the repository [37].

After the implementation was completed, faults seeding was done. Faults were seeded manually using a mutation testing technique mentioned by Jia and Harman [40] and Offutt, Lee [41]. In this experiment, three types of mutation were used, namely value mutation, decision mutation and statement mutation [42]. The total number of faults seeded for each web app is shown in the results section. To mitigate the bias of manually seeding the mutants, the line numbers of the possible locations for seeding mutants were randomly generated. Also, in this experiment, a modification represented a fault which meant more modifications created more faults. This approach seems impractical because, realistically, not all modifications done introduce faults. However, this approach was used because it portrayed the assumption that more modifications mean a higher possibility of introducing faults based on the study from Hassan [43]. All the web apps source code and the seeded faults information are available in the repository [37].

Lastly, the tests were run and their verdicts recorded automatically by a test execution environment in Eclipse IDE. For every fault seeded into the SUT, the test suite was executed to the faulty SUT to reveal which tests passed or failed. In MBT, when a test is run and the actual SUT does not perform exactly as expected by the test case, the test case will fail which shows that one or more faults are revealed in the SUT. During the execution, a table of fault matrices for each web app was completed that tabulated which test cases detected which faults. These tables are presented afterwards.

To perform prioritization, the abstract test cases were utilized because the modified transitions and states had to be identified and marked in each test case. To do this, faults injection information was used to pinpoint which parts of the coding that were seeded with faults. Then, the associated transitions or states of that part of coding in the test cases were identified and marked as modified transitions or states. Essentially, abstract test cases and modified transition information were the two inputs needed by the MB-TCP approaches to perform prioritization. Table 2 shows the modified transitions and states identified for the Online Jewellery Store

**Table 2. Modified transitions and states for the Online Jewellery Store.**

| Modified Transition | Transition Score, $ScT(T_j)$ |
|---|---|
| v_OrderHistoryPage | 1 |
| e_CompleteRegistration | 1 |
| e_CompleteCheckout | 4 |
| v_ViewProductPage | 2 |

FSM model. The transition score info was used in SESOC prioritization. Tables for the remaining web apps are available in the repository [37].

Lastly, to get the prioritized test suite, the table of fault matrices and MB-TCP approaches were utilized. Then, the ordering of test cases from each approach and the fault matrices were used to calculate the APFD values. Tables 3–5 tabulate the fault matrices for all three web apps. The rows show the faults while the columns show the test cases. "X" means that the test case failed or in other words, it detected the fault while "✓" means vice versa. The APFD results are shown in Section 6.

## 5.5 Threats to validity

The first issue that can affect the conclusion validity is the size of the sample. This experiment only made use of three web apps where the number also acted as the sample size. This can possibly affect the significance test result because the probability value could be higher than 0.05 significance level, thus making it impossible to reject the null hypothesis with strong evidence. However, based on the observation of the experiment result and the mean rank from Kruskal-Wallis H test in Section 6, SESOC is clearly more superior than other benchmark approaches for all web apps. This shows that if the sample size was increased, significant evidence that SESOC outperformed other benchmark approaches can be achieved. In addition, the statistical test used in the hypothesis testing was Kruskal-Wallis H. This is a type of non-parametric tests. It has a quite low statistical power so there are risks that an erroneous conclusion could be made. To reduce these risks, it was ensured that all the assumptions required to run this test have been adhered accordingly.

Next is the issue that can affect the internal validity of this experiment. The execution of this experiment required the assistance of several necessary tools and the implementation of some required processes. These could add variabilities to the result where other unintended independent variables might be affecting the dependent variable. To minimize these threats, it was ensured that the selected tools were appropriate in this experiment so that valid outcomes were produced. Also, the processes of MBT and prioritization were done carefully to prevent them from affecting the result. In addition, the constructed FSM models that were used to generate test cases were considered as correct in the experiment. In reality, it is actually possible

**Table 3. Fault matrices for Online Jewellery Shopping.**

| Fault | Test case | | | | | | | |
|---|---|---|---|---|---|---|---|---|
| | 1 | 2 | 3 | 4 | 5 | 6 | 7 | 8 |
| 1 | ✓ | X | ✓ | ✓ | ✓ | ✓ | ✓ | ✓ |
| 2 | ✓ | ✓ | X | ✓ | ✓ | ✓ | ✓ | ✓ |
| 3 | ✓ | X | ✓ | ✓ | X | ✓ | ✓ | ✓ |
| 4 | ✓ | X | ✓ | ✓ | X | X | X | X |
| 5 | ✓ | X | ✓ | ✓ | X | X | X | X |

**Table 4. Fault matrices for Car Rental System.**

| Fault | Test case | | | | | | | | | | | |
|---|---|---|---|---|---|---|---|---|---|---|---|---|
| | 1 | 2 | 3 | 4 | 5 | 6 | 7 | 8 | 9 | 10 | 11 | 12 |
| 1 | ✓ | ✓ | X | ✓ | ✓ | ✓ | ✓ | ✓ | ✓ | ✓ | ✓ | ✓ |
| 2 | ✓ | ✓ | ✓ | ✓ | ✓ | X | ✓ | ✓ | ✓ | ✓ | ✓ | ✓ |
| 3 | ✓ | X | ✓ | X | X | ✓ | ✓ | ✓ | X | ✓ | ✓ | X |
| 4 | ✓ | X | ✓ | ✓ | ✓ | ✓ | ✓ | ✓ | ✓ | ✓ | ✓ | ✓ |
| 5 | ✓ | ✓ | ✓ | X | X | ✓ | ✓ | X | X | ✓ | ✓ | X |
| 6 | ✓ | ✓ | ✓ | ✓ | X | ✓ | ✓ | ✓ | ✓ | ✓ | ✓ | ✓ |
| 7 | ✓ | ✓ | X | ✓ | ✓ | ✓ | ✓ | ✓ | ✓ | ✓ | ✓ | ✓ |

that the model itself was incorrectly designed. As a consequence, the generated test cases will be different and the prioritization process will be affected. In order to reduce the effect of this threat, the generated test cases were traced back to the requirements of the system to ensure full coverage. Full coverage means that there is at least one test case that tests a requirement and all requirements are tested by the test suite.

Regarding threats to construct validity, the issue is related to the dependent variable which is the APFD. In the experiment, it is stated that the experiment evaluated the approaches with respect to their effectiveness in prioritizing fault detecting tests so APFD was used. However, AFPD does not consider the cost of tests and severity of faults. In theory, when it comes to the effectiveness of fault detection, the cost of tests and severity of faults should also be taken into consideration. Therefore, the utilization of APFD could affect the construct validity of the experiment. This issue will be added in future work to utilize other metrics that can address the cost of tests and severity of faults such as Cost-cognizant Average Percentage of Faults Detected [44].

Lastly is to address the threats regarding external validity. The first issue is the web apps used in this experiment. They are open source web apps that are not in actual commercial use. Therefore, they might not represent web apps from real-world industries. To mitigate this threat, web apps that are similar and reflect those web apps from actual industrial were used. Another treats to external validity is related to the faults seeded. In the experiment, faults were

**Table 5. Fault matrices for Blood Bank Management System.**

| Fault | Test case | | | | | | | | | | | | | | | | | | | | | | | | |
|---|---|---|---|---|---|---|---|---|---|---|---|---|---|---|---|---|---|---|---|---|---|---|---|---|---|
| | 1 | 2 | 3 | 4 | 5 | 6 | 7 | 8 | 9 | 10 | 11 | 12 | 13 | 14 | 15 | 16 | 17 | 18 | 19 | 20 | 21 | 22 | 23 | 24 | 25 |
| 1 | ✓ | ✓ | ✓ | ✓ | ✓ | ✓ | ✓ | ✓ | ✓ | ✓ | ✓ | ✓ | ✓ | ✓ | ✓ | ✓ | ✓ | ✓ | ✓ | ✓ | X | ✓ | ✓ | ✓ | ✓ |
| 2 | ✓ | ✓ | ✓ | ✓ | ✓ | ✓ | X | ✓ | ✓ | ✓ | ✓ | ✓ | ✓ | ✓ | ✓ | ✓ | ✓ | ✓ | ✓ | ✓ | ✓ | ✓ | ✓ | ✓ | ✓ |
| 3 | ✓ | ✓ | X | ✓ | ✓ | ✓ | ✓ | ✓ | ✓ | ✓ | ✓ | ✓ | ✓ | ✓ | ✓ | ✓ | ✓ | ✓ | ✓ | ✓ | ✓ | ✓ | ✓ | ✓ | ✓ |
| 4 | ✓ | ✓ | ✓ | ✓ | ✓ | ✓ | ✓ | ✓ | ✓ | ✓ | ✓ | ✓ | X | ✓ | ✓ | ✓ | ✓ | ✓ | ✓ | ✓ | ✓ | ✓ | ✓ | ✓ | ✓ |
| 5 | ✓ | ✓ | ✓ | ✓ | ✓ | ✓ | X | ✓ | ✓ | ✓ | ✓ | ✓ | ✓ | ✓ | ✓ | ✓ | ✓ | ✓ | ✓ | ✓ | ✓ | ✓ | ✓ | ✓ | ✓ |
| 6 | ✓ | X | ✓ | ✓ | ✓ | ✓ | ✓ | ✓ | ✓ | ✓ | ✓ | ✓ | ✓ | ✓ | ✓ | ✓ | ✓ | ✓ | ✓ | ✓ | ✓ | ✓ | ✓ | ✓ | ✓ |
| 7 | ✓ | ✓ | ✓ | ✓ | ✓ | ✓ | ✓ | ✓ | ✓ | X | X | X | X | ✓ | ✓ | X | X | ✓ | ✓ | X | X | X | ✓ | ✓ | ✓ |
| 8 | ✓ | ✓ | X | X | ✓ | ✓ | ✓ | X | X | ✓ | X | ✓ | ✓ | ✓ | ✓ | ✓ | X | ✓ | ✓ | X | X | ✓ | X | X | ✓ |
| 9 | ✓ | X | ✓ | ✓ | ✓ | ✓ | X | ✓ | ✓ | ✓ | X | ✓ | ✓ | ✓ | ✓ | ✓ | ✓ | ✓ | X | X | ✓ | ✓ | ✓ | X | ✓ |
| 10 | ✓ | X | ✓ | ✓ | ✓ | ✓ | ✓ | ✓ | ✓ | ✓ | ✓ | ✓ | ✓ | ✓ | ✓ | ✓ | ✓ | ✓ | ✓ | ✓ | ✓ | ✓ | ✓ | ✓ | ✓ |
| 11 | ✓ | ✓ | ✓ | ✓ | ✓ | ✓ | ✓ | ✓ | ✓ | ✓ | ✓ | ✓ | ✓ | ✓ | ✓ | ✓ | ✓ | ✓ | ✓ | ✓ | X | ✓ | ✓ | ✓ | ✓ |
| 12 | ✓ | ✓ | ✓ | ✓ | ✓ | ✓ | ✓ | ✓ | ✓ | X | X | X | ✓ | ✓ | ✓ | ✓ | ✓ | ✓ | X | ✓ | ✓ | X | X | X | ✓ |
| 13 | ✓ | ✓ | ✓ | ✓ | ✓ | ✓ | X | ✓ | ✓ | ✓ | ✓ | ✓ | ✓ | ✓ | ✓ | ✓ | ✓ | ✓ | ✓ | ✓ | ✓ | ✓ | ✓ | ✓ | ✓ |

**Table 6. The number of possible orderings with their APFD values for Online Jewellery Store.**

| Approach | Number of Possible Orderings | Highest APFD (chances %) | Lowest APFD (chances) | APFD Mean |
|----------|------------------------------|--------------------------|-----------------------|-----------|
| STP | 720 | 0.9125 (3.33) | 0.6375 (2.5) | 0.7708 |
| BCP | 24 | 0.8875 (25.0) | 0.8125 (25.0) | 0.8500 |
| RCP | 24 | 0.8875 (25.0) | 0.8125 (25.0) | 0.8500 |
| BFP | 12 | 0.8375 (50.0) | 0.8125 (50.0) | 0.8250 |
| RFP | 12 | 0.8875 (50.0) | 0.8375 (50.0) | 0.8625 |
| SESOC | 24 | 0.9125 (100) | 0.9125 (100) | 0.9125 |

seeded into the SUT to create failed test cases. In real life industrial environment, it is obvious that testing is done to find actual faults in the SUT and not seeded faults. This issue will be mentioned in future work to obtain and utilize systems with actual faults for the experimentation so that it will reflect an actual industrial environment.

## 6.0 Results

Tables 6–8 tabulate the number of possible orderings, the highest and lowest APFD values and the mean APFD values from the three web apps. The first column contains all the associated approaches. The second column shows the number of possible orderings that can be generated from each approach. The third and fourth columns exhibit the highest and lowest APFD values from all the possible orderings of each approach and the chances of getting those values. Lastly, the fifth column displays the mean APFD of all values from the possible orderings of each approach. Higher APFD value means better prioritization result. All the raw data of possible orderings with their respective APFD values are available in the repository [37].

Several interesting observations can be pointed out from these tables. To discuss the number of possible orderings, the STP approach has the highest number of possible orderings for all case studies. For the highest APFD column, STP and SESOC both generated the highest APFD values for Online Jewellery Shopping and Blood Bank Management System with the values of 0.9125 and 0.9154 respectively. In addition, it can be observed that the highest APFD and lowest APFD values of SESOC for Blood Bank Management System case study are different while for Online Jewellery Shopping and Car Rental System case study, they are the same. This occurrence is interpreted in the discussion of the results in Section 7.

For Car Rental System, STP and RCP approaches have the highest APFD with the value of 0.8750. However, for chances percentage of highest APFD, SESOC outperforms all the other approaches with the values of 100%, 100% and 25% for Online Jewellery Shopping, Car Rental System and Blood Bank Management System respectively. For lowest APFD column, the STP approach generated the lowest APFD values for all dataset with the values of 0.6375, 0.6012 and 0.5369 respectively. Lastly, for the APFD Mean column, SESOC surpasses all other approaches with the highest APFD Mean value.

**Table 7. The number of possible orderings with their APFD values for Car Rental System.**

| Approach | Number of Possible Orderings | Highest APFD (chances %) | Lowest APFD (chances) | APFD Mean |
|----------|------------------------------|--------------------------|-----------------------|-----------|
| STP | 40320 | 0.8750 (0.12) | 0.6012 (0.09) | 0.7381 |
| BCP | 240 | 0.8393 (2.5) | 0.7083 (2.5) | 0.7738 |
| RCP | 240 | 0.8750 (2.5) | 0.6726 (2.5) | 0.7655 |
| BFP | 4 | 0.6964 (25.0) | 0.6726 (25.0) | 0.6845 |
| RFP | 4 | 0.6964 (25.0) | 0.6488 (25.0) | 0.6726 |
| SESOC | 24 | 0.8631 (100) | 0.8631 (100) | 0.8631 |

**Table 8. The number of possible orderings with their APFD values for Blood Bank Management System.**

| Approach | Number of Possible Orderings | Highest APFD (chances %) | Lowest APFD (chances) | APFD Mean |
|---|---|---|---|---|
| STP | 484248 | 0.9154 (0.02) | 0.5369 (0.004) | 0.7292 |
| BCP | 40320 | 0.8538 (0.05) | 0.74 (0.07) | 0.7959 |
| RCP | 40320 | 0.8815 (0.61) | 0.5953 (0.07) | 0.7144 |
| BFP | 40320 | 0.8477 (0.06) | 0.7338 (0.01) | 0.7918 |
| RFP | 40320 | 0.8477 (0.06) | 0.5677 (0.005) | 0.6979 |
| SESOC | 480 | 0.9154 (25.0) | 0.9092 (25.0) | 0.9123 |

Fig 7 represents the boxplots of APFD values for all possible orderings from each approach. Boxplot was used because it can visualize the dispersion and skewness of data well. The x-axis represents the web app while the y-axis represents the APFD value. The "X" mark in each box-plot shows the mean value. The boxplots were clustered into three groups based on web apps.

From the boxplot, STP has the biggest spread while SESOC has the smallest spread. Even though STP generated orderings with highest APFD values for Online Car Rental, SESOC has a smaller spread compared to STP for all web apps. This show that the APFD values from all possible orderings generated from SESOC are more consistent which is better because the chances of getting those high values are higher compared to STP.

Figs 8–10 show the line graphs of fault detection rate with APFD value for each web app respectively. The purpose of using the line graph is to visualize how rapid a prioritized test suite that is generated from an approach can detect all the seeded faults. The title for each graph is the approach's abbreviation with the APFD value of a selected ordering that is the same or nearest to the mean APFD from all the possible orderings. The x-axis represents the percentage of test suite coverage while the y-axis represents the percentage of faults detected.

From these graphs, a noteworthy observation can be pointed out where line graphs of SESOC for all datasets show the fastest 100 percent detection of faults. At first, it might seem in these graphs that if 100 percent detection of faults can be achieved with lesser test suite coverage, the APFD value will be higher. However, this is not always true because it can be observed that, for example, in STP and BFP graphs for Online Jewellery Shopping, both have the same test suite coverage for 100 percent detection of faults but their APFD values are different. This circumstance is further discussed in Section 7.

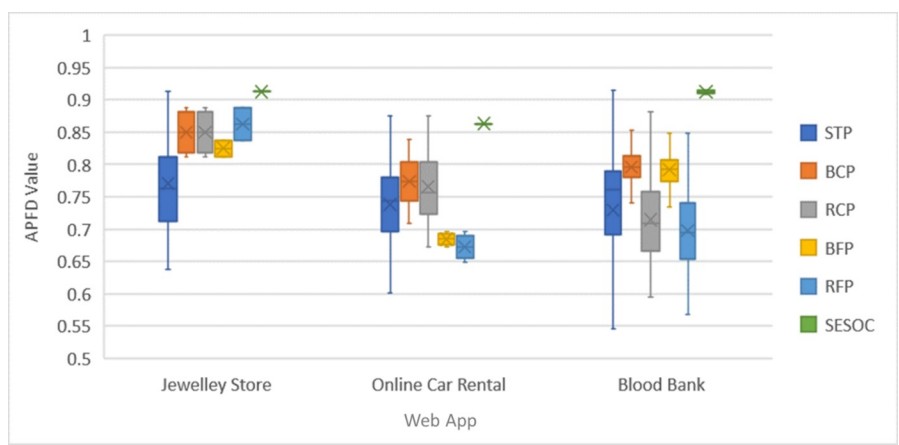

**Fig 7. Boxplots of APFD for all possible ordering from each approach.**

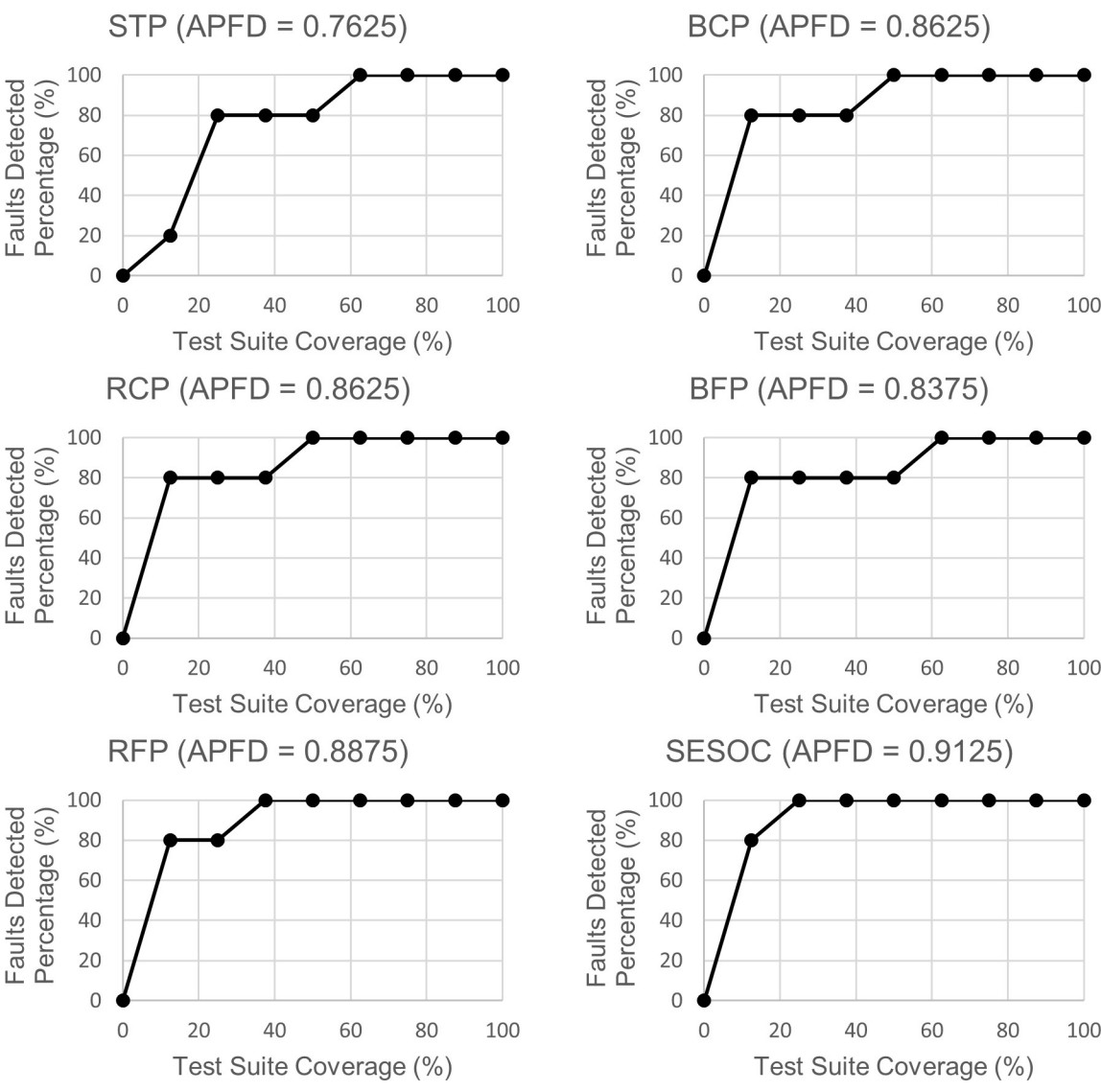

**Fig 8. Line graphs of faults detection rate for Online Jewellery Shopping.**

For hypothesis testing in this research, only the mean APFD values of each approach for all web apps were used as shown in Tables 6–8. All the calculations in the hypothesis testing were done using IBM SPSS Version 24. In this experiment, SESOC was compared between the existing MB-TCP approaches using FSM to determine whether it can outperform them. The null hypothesis is that there are no differences between the APFD values from the existing MB-TCP approaches using FSM and SESOC while the alternative hypothesis is vice versa. The initial plan was to utilize the one-way analysis of variance (ANOVA) but several assumptions were not met, one of them being the assumption that the dependent variable should be roughly normally distributed for each treatment of the independent variable. Table 9 shows the test for normality using the Shapiro-Wilk test. SESOC has a significance value that is lower than 0.05 (0.007) which shows that its distribution is not normal.

Therefore, the Kruskal-Wallis H test was used instead. Kruskal-Wallis H test is a rank-based non-parametric test that is used to conclude whether there are any statistical differences

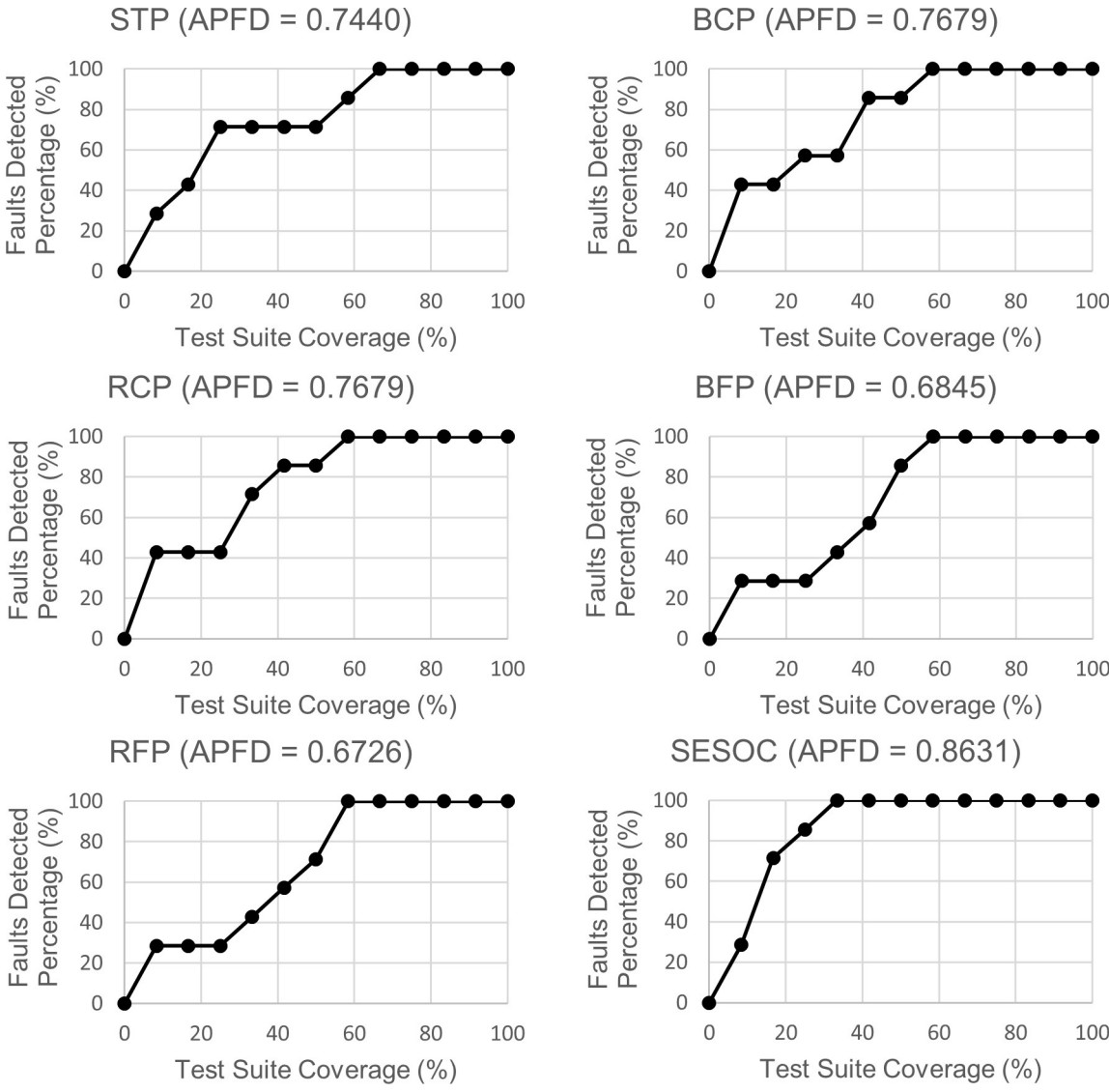

**Fig 9. Line graphs of faults detection rate for Car Rental System.**

between two or more treatments of an independent variable on an ordinal or continuous dependent variable. This test is considered as the alternative to one-way ANOVA in cases where ANOVA assumptions are not met. All the assumptions needed to run this test have also been adhered. Table 10 shows the test result using Kruskal-Wallis H for APFD value. The conclusion from this hypothesis testing is elaborated in Section 7.

## 7.0 Discussion

The first interpretation is that STP has the possibility to generate orderings that yield both highest and lowest APFD values. This can be observed in Tables 6–8 and the boxplots in Fig 7. The number of possible orderings column for STP in the tables prove that many possible orderings can be generated using STP. This happens because STP prioritization criteria only select test cases that traverse modified transitions to be prioritized while those that do not traverse modified transition are assigned last in the prioritized test suite. Thus, there will be many

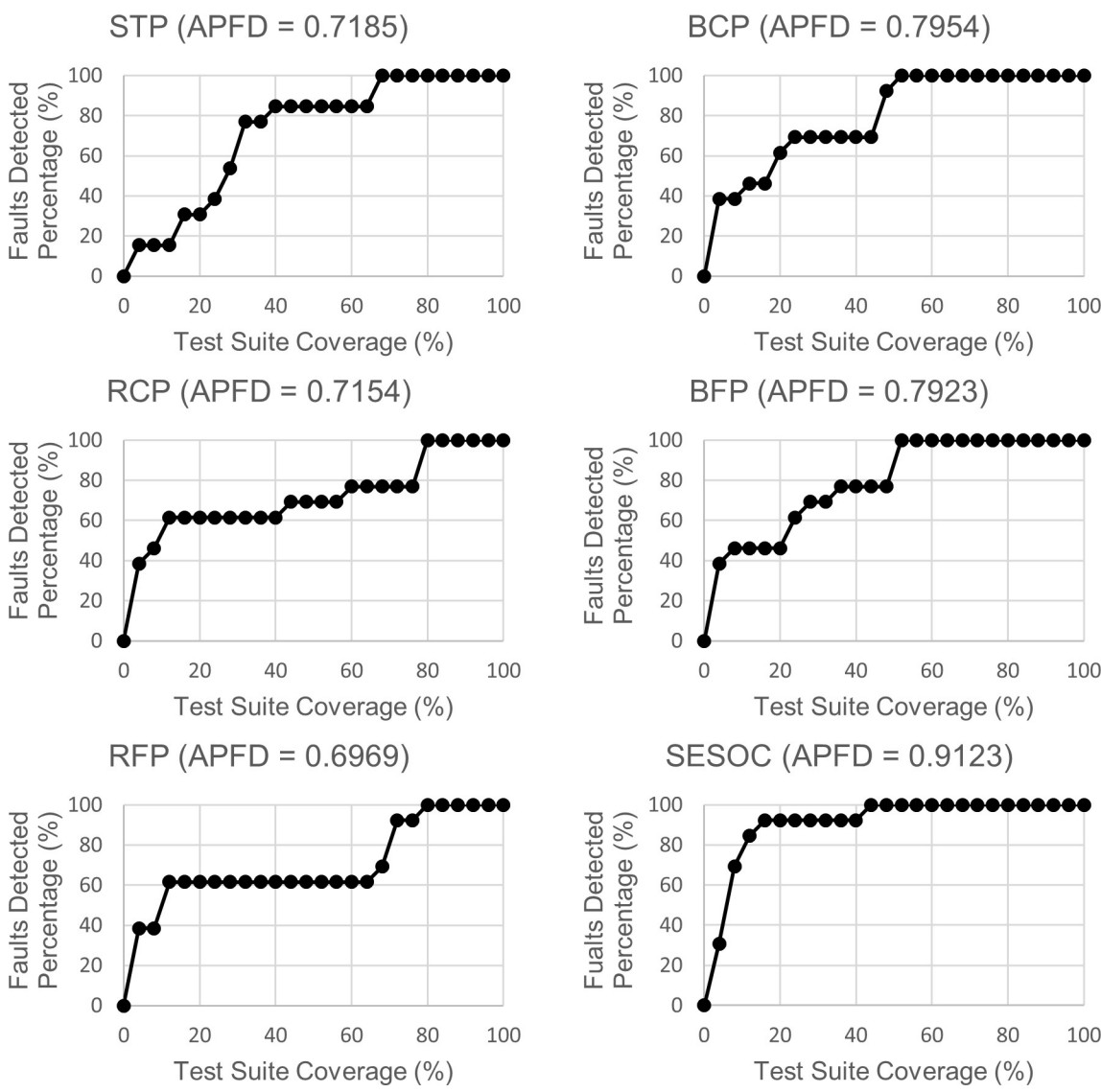

**Fig 10. Line graphs of faults detection rate for Blood Bank Management System.**

possible orderings that can be generated by STP which can include the best and the worst prioritization. Worst prioritization here only considers for the test cases with modified transitions. If test cases with no modified transition are included, the prioritization could be much

**Table 9. Test for normality using Shapiro-Wilk test.**

| Approach | Degree of Freedom | Significance |
|---|---|---|
| STP | 3 | 0.391 |
| BCP | 3 | 0.546 |
| RCP | 3 | 0.730 |
| BFP | 3 | 0.435 |
| RFP | 3 | 0.235 |
| SESOC | 3 | 0.007 |

**Table 10. Test result using Kruskal-Wallis H for APFD value.**

| Approach | Sample Size | Mean Rank | Significance |
|----------|-------------|-----------|--------------|
| STP | 3 | 6.33 | 0.119 |
| BCP | 3 | 11.17 | |
| RCP | 3 | 8.17 | |
| BFP | 3 | 8.00 | |
| RFP | 3 | 6.33 | |
| SESOC | 3 | 17.00 | |

worse. Therefore, STP approach can be utilized to indicate the worst prioritization and the best prioritization to be compared with other approaches.

In addition, it is obvious in Tables 6–8 that SESOC outperforms all other approaches in terms of APFD value. Plus, the chances for SESOC to obtain those high APFD values are also very convincing. Unfortunately, in Table 7, it seems that the highest APFD value is achieved by STP and RCP, not SESOC with a difference of 0.0119. However, if the chances are considered, SESOC has a higher chance of 100 percent to obtain the 0.8631 than STP and RCP with the chance of 0.12 and 2.5 percent respectively to obtain the 0.8750. This shows that SESOC is more likely to generate high APFD value than STP if the same number of possible orderings are to be generated from both approaches.

Also, it is mentioned earlier during the analysis of Figs 8–10 that it is not true that if 100 percent detection of faults can be achieved with lesser test suite coverage, the APFD value will be higher. This is because some pairs have the same test suite coverage for 100 percent detection of faults but their APFD values are different. The reason for this is because the calculation of APFD takes into consideration the number of test cases required until a fault is revealed, and this is done for all faults. This shows that it is not the test suite coverage to achieve 100 percent detection of faults that is considered when calculating APFD, but the test suite coverage to reveal each fault which is actually utilized. To prove this, it can be observed that in STP and BFP graphs for Online Jewellery Shopping, after the first test case is executed (the second dot in the line), BFP already achieves 80 percent detection of faults while STP just only achieves 20 percent detection of faults. This explains why BFP has higher APFD value than STP.

Furthermore, it is noticed that the highest APFD and lowest APFD values of SESOC for Blood Bank Management System case study are different while for Online Jewellery Shopping and Car Rental System, they are the same. This happens because the test suite size is small for Online Jewellery Shopping and Car Rental System; 8 and 12 test cases to be exact. Because of that, even though the selection of test cases can vary and different orderings might be produced, their APFD values are still the same. On the other hand, the test suite size of Blood Bank Management System is larger with 25 test cases so different possible orderings that could satisfy the conditions of SESOC will produce different APFD values. It can be concluded here that the size of the web app also affects the prioritization result obtained. When the web app size increases, the test suite size also increases, and the APFD values for all possible orderings generated from SESOC started to spread more as can be seen from the boxplot in Fig 7.

From the hypothesis testing, the Kruskal-Wallis H test showed that there is a weak evidence of a statistical difference in APFD value among the different approaches, $p = 0.119$, with a mean rank APFD value of 6.33 for STP, 11.17 for BCP, 8.17 for RCP, 8.00 for BFP, 6.33 for RFP and 17.00 for SESOC. Therefore, the null hypothesis cannot be rejected with significant evidence and must be retained. From the mean rank, SESOC has the highest rank which showed that the proposed approach is the most effective among other approaches. Unfortunately, the differences between all approaches are not significant because the significance value

is higher than 0.01, 0.05 and 0.1 significance levels. One of the reasons is because of the sample size, which in this case is the number of web apps, is too small so the data cannot supply enough evidence that the null hypothesis is false. Therefore, post hoc test to determine which of these approaches differ from each other cannot be run.

## 8.0 Conclusion

This study proposes an MB-TCP approach using FSM called SESOC. A brief description of the related subjects is also provided that include MBT, FSM and MB-TCP. To identify the gaps in the existing approaches, several related works in MB-TCP are reviewed. Several existing approaches are also used as the theoretical basis or foundation for SESOC. Then, the proposed approach is presented that aims at addressing the limitations found in the related works while improving the effectiveness of early fault detection during testing. A detailed experiment is conducted to evaluate and compare the effectiveness of early fault detection of SESOC with the existing approaches in the literature. The results obtained showed that SESOC outperformed the other approaches in terms of early fault detection. Nevertheless, this research is still far from perfection.

To improve this research more in the future, some crucial recommendations are suggested. First is to increase the sample size which is the number of web apps used so that stronger evidence can be obtained to reject the null hypothesis. The future plan also includes benchmarking SESOC with approaches from other categories like machine learning-based TCP or test case generation. This is so that SESOC effectiveness as an MBT approach can be further evaluated, not just regarding fault detection capability, but also in terms of execution cost of the approach itself. Finally is to consider the utilization of other metrics that can address the cost of tests and severity of faults such as Cost-cognizant Average Percentage of Faults Detected to strengthen the construct validity regarding the effectiveness in prioritizing faults detecting tests.

## Author Contributions

**Conceptualization:** Muhammad Luqman Mohd-Shafie, Wan Mohd Nasir Wan-Kadir.

**Data curation:** Muhammad Luqman Mohd-Shafie.

**Formal analysis:** Muhammad Luqman Mohd-Shafie.

**Funding acquisition:** Wan Mohd Nasir Wan-Kadir.

**Investigation:** Muhammad Luqman Mohd-Shafie.

**Methodology:** Muhammad Luqman Mohd-Shafie, Wan Mohd Nasir Wan-Kadir.

**Project administration:** Muhammad Luqman Mohd-Shafie, Wan Mohd Nasir Wan-Kadir.

**Resources:** Muhammad Luqman Mohd-Shafie, Mohd Adham Isa.

**Software:** Muhammad Luqman Mohd-Shafie.

**Supervision:** Wan Mohd Nasir Wan-Kadir.

**Validation:** Muhammad Luqman Mohd-Shafie, Wan Mohd Nasir Wan-Kadir, Muhammad Khatibsyarbini, Mohd Adham Isa.

**Visualization:** Muhammad Luqman Mohd-Shafie.

**Writing – original draft:** Muhammad Luqman Mohd-Shafie.

**Writing – review & editing:** Muhammad Luqman Mohd-Shafie, Wan Mohd Nasir Wan-Kadir, Muhammad Khatibsyarbini, Mohd Adham Isa.

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
