## [Decision Letter · Decision Letter 0]

20 Nov 2019

PONE-D-19-25986

Model-based test case prioritization using selective and even-spread count-based methods with scrutinized ordering criterion

PLOS ONE

Dear Dr. Wan-Kadir,

Thank you for submitting your manuscript to PLOS ONE. After careful consideration, we feel that it has merit but does not fully meet PLOS ONE’s publication criteria as it currently stands. Therefore, we invite you to submit a revised version of the manuscript that addresses the points raised during the review process.

We would appreciate receiving your revised manuscript by Jan 04 2020 11:59PM. To enhance the reproducibility of your results, we recommend that if applicable you deposit your laboratory protocols in protocols.io, where a protocol can be assigned its own identifier (DOI) such that it can be cited independently in the future. For instructions see: http://journals.plos.org/plosone/s/submission-guidelines#loc-laboratory-protocols

We look forward to receiving your revised manuscript.

Kind regards,

Seyed Reza Shahamiri

Academic Editor

PLOS ONE

Journal Requirements:

2. Please note that PLOS ONE has specific guidelines on software sharing (http://journals.plos.org/plosone/s/materials-and-software-sharing#loc-sharing-software) for manuscripts whose main purpose is the description of a new software or software package. In this case, new software must conform to the Open Source Definition (https://opensource.org/docs/osd) and be deposited in an open software archive. Please see http://journals.plos.org/plosone/s/materials-and-software-sharing#loc-depositing-software for more information on depositing your software.

3. We note you have included a table to which you do not refer in the text of your manuscript. Please ensure that you refer to Table 2 in your text; if accepted, production will need this reference to link the reader to the Table.

Additional Editor Comments:

• One of my observation is the problem statement and emphasis on regression testing execution costs, and the claim that 80% of the testing costs are spent on reg. testing execution. With modern unit test frameworks, the execution cost of regression testing doesn’t seem to be an issue anymore as the framework simply re-runs the predefined tests. It’s better that the authors further elaborate/revise the problem statement.

• There are some inconsistencies between the reference styles used. For example, page 3 last paragraph, “Yoo and Harman (1)…” while others use []. Please fix this issue.

• Please improve the English.

• How does the proposed approach compared with machine learning based test case generation/prioritizations technique? It would be good if the paper addresses this.

• How does the proposed approach address the oracle problem?

• I’m not sure of whether the authors have made all the data available, or provided enough detail for another researcher to reproduce the findings. Can you please clarify?

• I noted that the validation of the method presented in this manuscript might be incomplete. Whilst the performance has been compared to other existing methods, I’m unsure whether the methods chosen for comparison are state-of-the-art, and thus the results may not have been put in the right context. Please clarify this too.

• The structure of the paper is not adequate for a research paper. It seems the paper is written in a way a thesis should be written, especially section 5. I advise the authors to restructure the paper in a manner that is more acceptable for research papers rather than a thesis such as, Abstract, Introduction, Related work, Method, Experiments, Results, Discussion, Conclusion.

Reviewers' comments:

Reviewer's Responses to Questions

**Comments to the Author**

1. Is the manuscript technically sound, and do the data support the conclusions?

Reviewer #1: Partly

2. Has the statistical analysis been performed appropriately and rigorously? 

Reviewer #1: No

3. Have the authors made all data underlying the findings in their manuscript fully available?

Reviewer #1: No

4. Is the manuscript presented in an intelligible fashion and written in standard English?

Reviewer #1: No

5. Review Comments to the Author

Reviewer #1: 1. The objective of this research is to propose an MB-TCP approach that can improve the faults detection performance of regression testing. A detailed empirical study is conducted with the aims to evaluate and compare the performance of the proposed approach with the selected existing approaches from the literature using the average of the percentage of faults detected (APFD) metric. Three web applications were used. From the result obtained, the proposed approach yields the highest APFD values over other existing approaches which are 91%, 86% and 91% respectively for the three web applications.

2. Introduction section is very lengthy. Some information pertaining to the research is better to be discussed in section 4.

3. In am not sure what is the different between section 2 and 3. This is because both sections has similar contents. The background section should be more general, while the related work should cover the works that are applied in the current research

4. The heading of section 4 is proposed MB-TCP approach. It is confusing as prior to this, it was mentioned in section 3 that MB-TCP approach is an existing approach. To reduce the confusion, heading for section 4 is better as “proposed approach”

5. The authors did not explain why they have selected the three websites only. Justification needed.

6. The main issue that forms the basis for research is the high execution cost. However, the authors did not perform testing of their proposed method on the execution cost. This is the biggest limitation of this research. We cannot determine the effectiveness of the proposed approach in relation to the execution cost as it is major issue highlighted in the abstract.

7. The sample size is based on the number of websites. However, each website has different line of command. I think it is better to use line of code as research unit rather than the website itself.

8. The use of Shapiro–Wilk test for hypothesis testing is questionable. The null hypothesis is that there are no differences between the APFD values from the existing MB-TCP approaches using FSM and SESOC. However, Shapiro–Wilk test is to test the normality. The finding suggests that that SESOC is not normally distributed. It did not suggest that they are different. The authors should use ANOVA as described in section 4.

9. Figures are off poor quality and cannot be comprehended. Some figures are missing from the document.

6. PLOS authors have the option to publish the peer review history of their article (what does this mean?). If published, this will include your full peer review and any attached files.

Reviewer #1: Yes: Mumtaz Begum Mustafa, Ph.D.

---

## [Author Response · Author response to Decision Letter 0]

4 Jan 2020

Journal requirements: I have addressed all the journal requirements stated in the decision letter.

Academic editor: I have addressed all the academic editor comments stated in the decision letter.

Reviewer #1: I have addressed all the Reviewer #1 comments stated in the decision letter.

The details are presented in a table of corrections (with page and line number) uploaded as the response to reviewers file.

---

## [Decision Letter · Decision Letter 1]

4 Feb 2020

Model-based test case prioritization using selective and even-spread count-based methods with scrutinized ordering criterion

PONE-D-19-25986R1

Dear Dr. Wan-Kadir,

We are pleased to inform you that your manuscript has been judged scientifically suitable for publication and will be formally accepted for publication once it complies with all outstanding technical requirements.

With kind regards,

Seyed Reza Shahamiri

Academic Editor

PLOS ONE

Additional Editor Comments (optional):

Reviewers' comments:

Reviewer's Responses to Questions

**Comments to the Author**

1. If the authors have adequately addressed your comments raised in a previous round of review and you feel that this manuscript is now acceptable for publication, you may indicate that here to bypass the “Comments to the Author” section, enter your conflict of interest statement in the “Confidential to Editor” section, and submit your "Accept" recommendation.

Reviewer #1: All comments have been addressed

2. Is the manuscript technically sound, and do the data support the conclusions?

Reviewer #1: Yes

3. Has the statistical analysis been performed appropriately and rigorously? 

Reviewer #1: Yes

4. Have the authors made all data underlying the findings in their manuscript fully available?

Reviewer #1: Yes

5. Is the manuscript presented in an intelligible fashion and written in standard English?

Reviewer #1: Yes

6. Review Comments to the Author

Reviewer #1: (No Response)

7. PLOS authors have the option to publish the peer review history of their article (what does this mean?). If published, this will include your full peer review and any attached files.

Reviewer #1: Yes: Mumtaz Begum Mustafa

---

## [Editor Report · Acceptance letter]

6 Feb 2020

PONE-D-19-25986R1 

Model-based test case prioritization using selective and even-spread count-based methods with scrutinized ordering criterion 

Dear Dr. Wan-Kadir:

I am pleased to inform you that your manuscript has been deemed suitable for publication in PLOS ONE. Congratulations! Your manuscript is now with our production department. 

With kind regards,

on behalf of

Dr. Seyed Reza Shahamiri 

Academic Editor

PLOS ONE